# Adaptation of the Concept of Brain Reserve for the Prediction of Stroke Outcome: Proxies, Neural Mechanisms, and Significance for Research

**DOI:** 10.3390/brainsci14010077

**Published:** 2024-01-12

**Authors:** Roza M. Umarova, Laura Gallucci, Arsany Hakim, Roland Wiest, Urs Fischer, Marcel Arnold

**Affiliations:** 1Department of Neurology, University Hospital Inselspital, University of Bern, 3010 Bern, Switzerland; laura.gallucci@extern.insel.ch (L.G.); urs.fischer@insel.ch (U.F.); marcel.arnold@insel.ch (M.A.); 2Department of Neuroradiology, University Hospital Inselspital, University of Bern, 3010 Bern, Switzerland; arsany.hakim@insel.ch (A.H.); roland.wiest@insel.ch (R.W.); 3Department of Neurology, University Hospital Basel, University of Basel, 4003 Basel, Switzerland

**Keywords:** brain reserve, cognitive reserve, stroke outcome, prediction, brain atrophy, white matter hyperintensity, brain health

## Abstract

The prediction of stroke outcome is challenging due to the high inter-individual variability in stroke patients. We recently suggested the adaptation of the concept of brain reserve (BR) to improve the prediction of stroke outcome. This concept was initially developed alongside the one for the cognitive reserve for neurodegeneration and forms a valuable theoretical framework to capture high inter-individual variability in stroke patients. In the present work, we suggest and discuss (i) BR-proxies—quantitative brain characteristics at the time stroke occurs (e.g., brain volume, hippocampus volume), and (ii) proxies of brain pathology reducing BR (e.g., brain atrophy, severity of white matter hyperintensities), parameters easily available from a routine MRI examination that might improve the prediction of stroke outcome. Though the influence of these parameters on stroke outcome has been partly reported individually, their independent and combined impact is yet to be determined. Conceptually, BR is a continuous measure determining the amount of brain structure available to mitigate and compensate for stroke damage, thus reflecting individual differences in neural resources and a capacity to maintain performance and recover after stroke. We suggest that stroke outcome might be defined as an interaction between BR at the time stroke occurs and lesion load. BR in stroke can potentially be influenced, e.g., by modifying cardiovascular risk factors. In addition to the potential power of the BR concept in a mechanistic understanding of inter-individual variability in stroke outcome and establishing individualized therapeutic approaches, it might help to strengthen the synergy of preventive measures in stroke, neurodegeneration, and healthy aging.

## 1. Introduction

The prediction of stroke outcome remains challenging due to the high inter-individual variability in stroke patients [1]. This also includes the heterogeneous presence of brain pathology before having a stroke event. Nowadays, inter-individual variability in stroke patients is characterized mainly by age, vascular risk factors, and stroke severity. It is striking, however, how stroke severity measured via NIH Stroke Scale (NIHSS) reflects lesion load and predicts functional outcome [1]. At the same time, lesion size and anatomy also have a restricted explanatory power in the prediction of stroke outcome: even high-dimensional lesion feature analysis may explain only up to 40% of acute stroke severity [2]. In line with the point, even a lacunar stroke may lead to dementia in some patients, whereas in others, territorial ischemia does not lead to any apparent cognitive deficit [3]. Thus, improved models to better capture the inter-individual variability are needed. To explain the inter-individual variability in response to neurodegenerative pathology, the concepts of brain and cognitive reserve have been introduced. ‘Brain reserve’ (BR) is defined by quantitative brain parameters, operationalized through such proxies as brain size, hippocampal volume, or white matter microstructural characteristics before the pathology occurs [4]. This is considered to be a passive model, as an active influence on these parameters is impossible. ‘Cognitive reserve’ (CR) has been defined as the function of lifetime intellectual activities, which serve to shape network efficiency, processing capacity, and flexibility, with such proxies as level of education, occupational history, crystallized intelligence, socioeconomic status, etc. [4]. CR represents the active model, meaning that dynamic cognitive and underlying functional brain processes cope with brain changes or damage [5]. Altogether, both reserves can act as a moderator between pathology and clinical outcome, thus accounting for this discontinuity [6].

Recently, we suggested that the concepts of BR and CR, both established for neurodegeneration, might represent a valuable theoretical framework to explain patient’s inter-individual variability in stroke outcome and recovery [7]. Due to the distinct nature of the pathology and disease course—chronic-progressive versus acute event—the concepts of BR and CR should be adapted for stroke. The purpose of the present work is to adapt the BR concept and to suggest the proxies of BR as well as proxies of its pathology to predict stroke outcome. Though we showed that CR is to be considered in the prediction of stroke outcome [8,9], we do not discuss this concept in the present work for simplicity reasons. In line with the BR concept in neurodegeneration, we suggest that stroke outcome might be defined as an interaction between BR at the time stroke occurs and for lesion load.

## 2. Potential Proxies of Brain Reserve in Stroke Patients

The concept of BR has been established for neurodegenerative diseases with slow pathology accumulation and brain tissue loss. Correspondingly, those patients, who possess more brain tissue before the disease occurs, tolerate more pathology. Stroke is distinguished principally from neurodegeneration, as an acute pathology, after which the recovery process takes place. Thus, the structural “reserves” at the time stroke occurs are to be considered. Parameters reflecting in a sufficient way the inter-individual variability in quantitative brain characteristics at the time stroke occurs are potential candidates for proxies of BR.

The most widely used proxy of BR in neurodegeneration is the total intracranial volume (TIV). Although the measure includes the cerebrospinal fluid volume, it neatly represents the maximal brain volume along the lifespan. Patients with higher TIV might tolerate more neurodegeneration before the critical threshold of pathology is achieved and dementia occurs [10,11]. Similar results have been shown for multiple sclerosis [12]. For the prediction of stroke outcome, however, TIV may be unsuitable as a proxy for BR. By reflecting the brain size in the past, TIV is irrelevant to an individual’s response to stroke lesion and the recovery process. In contrast, brain volumetric characteristics at the time stroke occurs are potential candidates to represent BR in the prediction of stroke outcome (Figure 1). Until now, different structural brain characteristics, including those derived from advanced MRI imaging, have been reported to impact cognitive stroke outcome [13]. There is, however, increasing evidence of its potential impact on non-cognitive and functional outcome, namely regarding the severity of neurological deficits, disability, and dependency [7].

The impact on stroke outcome has been demonstrated for several quantitative brain characteristics easily available from the routine MRI examination (Figure 1). *Brain volume* impacts functional stroke outcome, as its independent predictor [14,15]. To date, whether the grey and white matter volumes or grey and white matter densities possess distinct and relevant impacts on stroke outcome have not been investigated. Several types of brain pathology may reduce BR and thus modify stroke outcome. One such proxy of brain pathology is *brain atrophy,* which represents the severity of brain volume loss at the time of stroke occurrence. Though it might correlate with brain volume, it represents, rather, the severity of neurodegenerative processes or accelerated aging and therefore possesses a distinct, brain-size-independent impact. For clinical decision-making, brain atrophy might be assessed, e.g., via ‘the global cerebral atrophy’ score [16]: by providing only a rough rating, this method is easily applicable for CT scans. For stroke research, the brain parenchymal fraction might be calculated as the individual index of brain atrophy (a ratio of brain parenchymal volume to the total intracranial volume). Volumetric analysis is also becoming available in the clinical routine, along with the high availability of the MRI examination in stroke patients. Thus, both proxies—brain volume and its atrophy—might capture different variances in the elderly population. Despite the application of different methods on brain atrophy assessment, several studies demonstrated its negative impact on stroke outcome and recovery trajectories [17,18,19,20,21]. The impact of brain atrophy in malignant stroke has been reported to be more complex, with a trend towards mortality reduction but a worsening of functional recovery [22].

*White matter hyperintensities* represent the proxy of white matter pathology [23] (Figure 1), which is assessed via Fazekas in the clinical routine praxis. According to a recent meta-analysis, the presence and severity of white matter hyperintensities are associated with worse stroke outcome [24], including outcome after minor stroke [25]. Mechanistically, white matter hyperintensities might be associated with blood–brain barrier dysfunction, impaired vasodilation, vessel stiffening, dysfunctional blood flow, interstitial fluid drainage, white matter rarefaction, local ischemia, inflammation, myelin damage, and secondary neurodegeneration [23]. All these parameters reduce BR. Advanced neuroimaging studies have demonstrated that some affected parameters of the “normal appearing white matter” might be also associated with worse stroke outcome [26,27]. Secondary neurodegeneration has been discussed as the main mediator of impact of white matter hyperintensities on cognition [28,29]. There are other features of white matter pathology—microbleeds, lacunes, perivascular spaces—which together with white matter hyperintensities are defined as small vessel disease. Though each of them was demonstrated to be associated with stroke outcome or post-stroke cognition [30], their independent impact on stroke outcome is still poorly investigated [31] and should be a focus of future research.

To the best of our knowledge, there are no studies assessing the effect of the medial temporal lobe volume or its atrophy on non-cognitive stroke outcome. The negative impact of hippocampus atrophy has been shown in regard to post-stroke cognition [32,33,34,35]. On the other hand, this parameter might be associated with the presence of Alzheimer pathology, making the causative inference complex. Being involved in the learning process, medial temporal lobe volume might moderate the effectiveness of rehabilitation and consequently stroke outcome. Silent stroke also reduces BR and is shown to be associated with worse cognitive performance independent of hippocampus volume [36,37]. The negative impact of previous silent stroke on stroke outcome has not been reported yet [38,39], though knowledge gaps in this field are reported [40]. Other proxies of BR might be driven from advanced neuroimaging methods such as diffusion imaging-derived white matter characteristics, or parameters of functional [41,42] or structural [43] brain connectivity. The measures of disrupted functional and structural connectivity predicted significant variance in cognitive and motor outcomes in stroke [44,45]. Diffusion-tensor parameters such as the fractional anisotropy or mean diffusivity of the specific fiber tracks have been shown to be helpful in the prediction of motor outcome [46,47]. The parameters driven from connectivity or DTI analysis are, however, not yet established in routine clinical practice due to insufficient robustness [48], and methodological limitations [49]. Moreover, it is yet to be investigated whether advanced neuroimaging parameters—such as connectivity or diffusion-tensor parameters—are superior to a comprehensive analysis of routine BR-proxies in their predictive power for stroke outcome.

Given that several proxies of BR and brain pathology exist, their combined impact should be addressed. This was undertaken in research on normal aging [50] and cognitive performance [51,52] with analyses of cortical thickness, brain atrophy, and white matter hyperintensities in one model. However, there are only a few studies that consider more than one parameter of brain pathology to predict stroke outcome. Recent studies analyzed the presence of white matter hyperintensities, brain atrophy, and old vascular lesions in a categorical way, and demonstrated their negative impact on stroke outcome [53,54]. In contrast, Bu et al. (2021) analyzed the independent impact of brain volume, its atrophy, and the severity of small vessel disease assessed in a quantitative way. From these features, only brain atrophy was associated with stroke outcome [17]. The features of brain pathology are, however, interrelated and are not strictly independent. In addition, the same proxy might represent the manifestation of different pathologies. For example, brain atrophy might be a manifestation of neurodegenerative pathology and severe small vessel disease. Thus, there are the following knowledge gaps that should be addressed in future studies: (i) What is the combined and independent impact of different types of brain pathology? (ii) Do the different types of brain pathology impact stroke outcome in an additive or over-additive way? (iii) Do advanced neuroimaging parameters (cortical thickness, fractional anisotropy, etc.) overperform parameters derived in the routine clinical praxis (global brain atrophy, white matter hyperintensities, etc.) in the explained variance of stroke outcome?

## 3. Neural Mechanisms of Impact of Brain Reserve and Its Pathology on Stroke Outcome

Stroke recovery is enabled by diffuse and redundant connectivity in the central nervous system and the ability of new structural and functional circuits to form through remapping between related cortical regions [55]. This ability is highly dependent on individual structural and volumetric brain characteristics available after a stroke event. Altogether, the suggested BR-proxies and proxies of its pathology characterize roughly the quantity and quality of brain tissue available after stroke damage (Figure 1). Proxies which characterize grey matter (e.g., cortical thickness, grey matter density) roughly reflect its microstructural characteristics: synaptic density, number of neurons, etc. On the network level, these proxies underpin the properties of cortical network cores. Similarly, the parameters of white matter and its pathology reflect white matter integrity and network connectivity; e.g., white matter hyperintensities have been shown to correlate negatively with brain network efficiency, network integration, and segregation (for a review, please see [56]). Due to this strong functional–structural interrelation, it is therefore difficult to segregate poor (micro)structural quantitative characteristics that compose BR according to the strict original BR definition in neurodegeneration [4] from BR-related functional network characteristics. Considering concepts of healthy aging in neuroscience [57], BR might mitigate stroke damage, first, by representing more neural resources: the more structure available, the less impact of stroke with the specific size and location (Figure 2) [4]. Second, BR might influence maintenance after stroke, namely neural mechanisms of repair and plasticity. Stroke leads to structural, recovery-relevant re-modeling even in unaffected hemispheres [58], and this process might be influenced by BR. Finally, BR might impact a network’s compensation, namely enhancing the recruitment of additional neural resources in response to relatively high task demand. This was shown for motor outcome in stroke patients: quantitative characteristics of interhemispheric structural connectivity correlated with motor recovery [59].

It is indisputable that the impact of stroke lesion on outcome is strongly associated with its location and size [2]. Severe stroke lesions—larger and strategically located ones—compromise more BR, as less brain tissue and fewer functional centers remain available, and therefore the neural plasticity might be more constrained [7]. Therefore, in addition to the simple linear effect of BR and lesion load on stroke outcome described, we have also suggested a more complex interaction effect between them [9]: we see stroke outcome as an interaction effect between lesion load and available BR at the time stroke occurs. This model explicitly explains the individual burden of stroke damage: Individuals with higher BR might tolerate more severe stroke damage—lesions with critical location and larger size—and still show a favorable stroke outcome. In contrast, patients with low BR might suffer poor stroke outcome despite small uncritical lesions (Figure 2). The systematic way, in which one should balance BR against lesion load and location is, however, to be further defined based on the multivariate big data.

## 4. Discussion

The consideration of BR has several advantages (Table 1). First, it might improve our understanding of the inter-individual variability in stroke outcome, including the prediction of long-term stroke outcome, which has been an under-developed research field until now. More specifically, the concept might help us to understand the individual burden of stroke damage by considering the interaction between an individual’s BR and lesion load. Therapeutic trials might also benefit from the consideration of BR-proxies due to better control for potential confounding factors, thus increasing studies’ validity and power. Second, instead of operating with the single proxies, it may be helpful to introduce the global, continuous measure of BR reflecting structural brain health, e.g., through the imaging-based “brain age” parameter [60]. Moreover, the concept suggests the multi-dimensional consideration of multiple proxies, e.g., protective BR-proxies, as well as proxies of different brain pathology, instead of the one-dimensional consideration of cardiovascular co-morbidities or single variables such as white matter hyperintensities or brain atrophy only. It might be more reasonable to look at such a multi-dimensional continuous evaluation of multiple proxies in relation to a given stroke lesion. For example, one patient might have white matter hyperintensities and brain atrophy, and another one none of these, but the former might suffer only lacunar lesion and the latter one territorial ischemia, and both demonstrate the same stroke outcome. Third, there is potential knowledge transfer: BR-proxies shown to be important in the manifestation of one pathology (e.g., neurodegenerative diseases) might also impact stroke outcome. Fourth, in contrast to neurodegeneration, where the BR concept operates with unmodifiable proxies (e.g., total intracranial volume), BR in the context of stroke outcome prediction might be modified along life span lines by primary (healthy lifestyle) or secondary (better control for cardiovascular risk factors) preventative measures [61]. Importantly, taking into account the effectiveness of a disease-modifying lifestyle in neurodegeneration and cardiovascular diseases, the recommendations for a healthy lifestyle become not disease-specific anymore, but rather universal, potentially improving stroke outcome by the moderation of BR. These points might help physicians to improve patients’ compliance and adherence. Finally, the concept of brain reserve might help to establish individualized therapeutic approaches, allowing the effective use of public health resources.

Nowadays, several other related concepts have been formulated and a clear distinction between them and the BR concept is warranted. One of the recently introduced concepts—‘brain resilience’—refers mainly to brain capacity to respond to stressors [62] and not to structural brain damage such as stroke. ‘Brain health’ is a widely used term that according to the recent definition of the World Health Organization includes factors such as emotional, psychological, and behavioral functioning to cope with life situations [63]. Thus, the ‘brain health’ concept operates with numerous interconnected structural, social, and psychological determinants, and goes far beyond the BR concept introduced. The term ‘brain health’ has been also used in the narrower meaning to characterize a pronouncement of cerebrovascular damage [64] or combined cerebrovascular and neurodegenerative damage [51], mainly to explain the variance in cognitive functioning after stroke. The term “brain frailty” is also used to describe the pre-existing signs of cerebrovascular disease. Both concepts, ‘brain health’ and ‘brain frailty’, operationalize the features centered on cerebrovascular pathology. Roughly comparable to the two concepts mentioned above, BR concept differs by (i) operationalizing protective brain characteristics, e.g., brain volume or cortical thickness, (ii) including all proxies of brain pathology and not only those of cerebrovascular pathology, (iii) considering lesion load to define the individual burden of stroke damage. We argue that the omission of characteristics of BR and the consideration of only proxies of pathology is insufficient to explain variability in stroke outcome. Nowadays, many other terms are introduced to describe the brain’s capacity to respond or adapt to brain damage. The main purpose of the present work, however, is not to introduce the additional term but rather (i) to motivate new discussions and research on the consideration of both the proxies of BR and of brain pathology, and (ii) to operate within these parameters in a continuous way to better capture and account for the variability in stroke outcome and to predict it.

## Figures and Tables

**Figure 1 brainsci-14-00077-f001:**
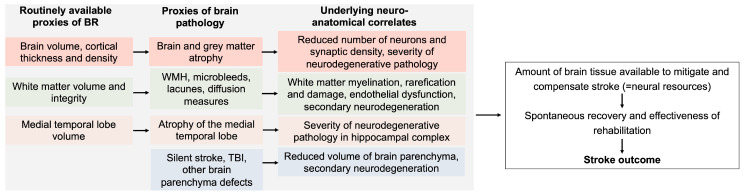
Proxies of brain reserve and brain pathology, and their mechanistic impact on stroke outcome. WMH—white matter hyperintensities; TBI—traumatic brain injury.

**Figure 2 brainsci-14-00077-f002:**
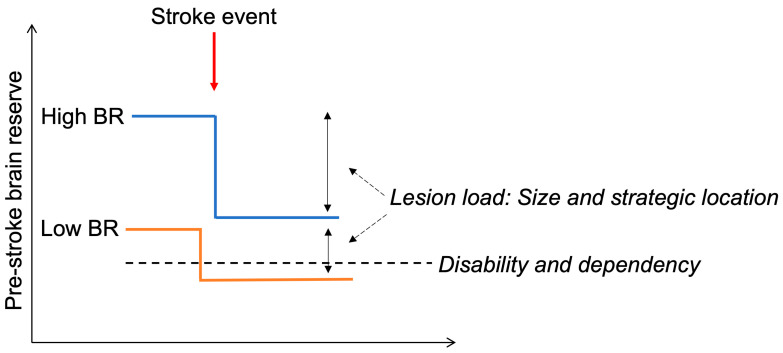
The level of pre-stroke brain reserve (BR) might impact functional stroke outcome and explain inter-individual variability. Patients with higher brain reserve (blue line) might maintain functional independence despite more severe stroke damage (e.g., larger and more critically located lesions), whereas patients with lower brain reserve (red line) might tolerate less stroke pathology and develop functional dependency and disability even after uncritical lesion.

**Table 1 brainsci-14-00077-t001:** Advantages and relevance of application of the concept of brain reserve in stroke.

Understanding of the inter-individual variability in stroke patients: - Understanding of individual stroke burden through consideration of both individual brain characteristics and stroke damage; - Consideration of potential confounders and adjustments for population differences in trials; - Prediction of (long-term) stroke outcome.
Development of a global measure for brain reserve: - Operationalization of BR as a *continuum* and not as a binary variable of the presence or absence of brain pathology; - Multi-dimensional consideration of BR, e.g., consideration of not only cerebrovascular pathology but also of protective BR-proxies, neurodegenerative, post-traumatic changes, etc.
Potential knowledge transfer: - Proxies reported to be important in the manifestation of one disease might also have an impact in stroke. Strengthen synergy in preventive interventions in neurodegeneration, stroke, and healthy aging.
Global impact of healthy and disease-modifying lifestyle to improve BR: - Primary (healthy lifestyle) and secondary (better control for cardiovascular risk factors) prevention.
Individualized therapeutic approaches

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
