# Peer review of "Adaptation of the Concept of Brain Reserve for the Prediction of Stroke Outcome: Proxies, Neural Mechanisms, and Significance for Research"

_brainsci, 2024, doi:10.3390/brainsci14010077_

Round 1
Reviewer 1 Report
Comments and Suggestions for Authors
Overall comment: I really appreciate the authors' intention to better utilize the concept of "effective reserve/brain reserve" for stroke. Given the nature of this manuscript as an opinion paper/commentary, there are several conceptual comments/concerns regarding this manuscript:
1. Throughout the manuscript, authors compared and contrasted neurodegenerative conditions versus stroke, which could lead to significant amount of confusion to readers. I would recommend authors revise part 2 of the manuscript and remove discussions regarding neurodegeneration.
2. It is unclear in terms of what aspects of "stroke" authors were referring to. The key proxies authors mentioned, for example, brain atrophy or white matter microstructural characteristics can function as indicator for different aspects of stroke. Are authors referring to stroke prediction, or recurrent stroke, or post-stroke recovery? It is really important if authors could define "stroke" clearly in the title and the aim of this manuscript.
3. The concept of "inter-individual variability" needs more elaboration. What kind of variability are we looking at? And are authors interested in intra-personal variability?
4. The biggest challenge for readers of this manuscript is that it was not clear what are the recommended adaption to the concept of "brain reserve". One example, in the Discussion section, authors mentioned: "It might be more reasonable to look at the overall measure of BR and proxies of pathology in relation to a given stroke lesion and not on single variables such as white matter hyperintensities or brain atrophy only." Readers were prompted to think about the definition of "overall measure of BR" which was not discussed in the manuscript.
Overall, authors need to revise the definition of the research question of interest, and clearly define "stroke" as well as the adapted concepts for BR.
Comments on the Quality of English Language
Consider revise the title and edit the grammar throughout the manuscripts.
Author Response
Overall comment: I really appreciate the authors' intention to better utilize the concept of "effective reserve/brain reserve" for stroke. Given the nature of this manuscript as an opinion paper/commentary, there are several conceptual comments/concerns regarding this manuscript:
We thank the Reviewer for the constructive critiques, comments and appreciation of our work.
- Throughout the manuscript, authors compared and contrasted neurodegenerative conditions versus stroke, which could lead to significant amount of confusion to readers. I would recommend authors revise part 2 of the manuscript and remove discussions regarding neurodegeneration.
Many thanks for the point. Indeed, there could be different approaches to discuss the concept of BR in the context of prediction of stroke outcome. However, if we remove the discussion regarding neurodegeneration, one may wonder what the differences between BR concept in prediction of stroke outcome and dementia are. We sought to avoid a deep discussion of BR concept in neurodegeneration. But we do believe that it is important to provide a context of BR concept in the field of neurodegeneration for readers. In addition, the Reviewer requires more details on “adaptation” of BR concept in the context of stroke outcome in his Point 4. We do appreciate the point to make the manuscript clearer for readers. We revised the manuscript in line with other points of the Reviewer and hope to improve the manuscript now.
- It is unclear in terms of what aspects of "stroke" authors were referring to. The key proxies authors mentioned, for example, brain atrophy or white matter microstructural characteristics can function as indicator for different aspects of stroke. Are authors referring to stroke prediction, or recurrent stroke, or post-stroke recovery? It is really important if authors could define "stroke" clearly in the title and the aim of this manuscript.
Many thanks for the point. We referred to stroke as an acute event according to the definition of WHO. In line with the Reviewer comment, we revised the manuscript and its title now and provided the context of “stroke” term clearer now.
- The concept of "inter-individual variability" needs more elaboration. What kind of variability are we looking at? And are authors interested in intra-personal variability?
We did not mention any concept of “inter-individual variability”. We rather refer to the inter-individual variability in stroke outcome and stroke recovery. We are very sorry for the confusion. We sought to use the term more precisely now. We were not interested in the intra-personal variability.
- The biggest challenge for readers of this manuscript is that it was not clear what are the recommended adaption to the concept of "brain reserve". One example, in the Discussion section, authors mentioned: "It might be more reasonable to look at the overall measure of BR and proxies of pathology in relation to a given stroke lesion and not on single variables such as white matter hyperintensities or brain atrophy only." Readers were prompted to think about the definition of "overall measure of BR" which was not discussed in the manuscript.
Overall, authors need to revise the definition of the research question of interest, and clearly define "stroke" as well as the adapted concepts for BR.
Many thanks for the point. Concerning the “adaptation” of BR concept, we emphasized that for example the widely used BR proxy in neurodegeneration - total intracranial volume - is not applicable to improve prediction of stroke outcome. Furthermore, in neurodegeneration, the proxies of pathology are not considered. We are sorry, if it was unclear in the previous version of the manuscript. We revied the manuscript to make it clearer now. The term “overall measure of BR and proxies of pathology” referred to the previous two sentences. We revised this part of the discussion now to avoid any misinterpretation.
In line with the point 2, we sought to refer to the term “stroke” more precisely now.

Reviewer 2 Report
Comments and Suggestions for Authors
“Adaptation of the concept of brain reserve for stroke: proxies, neural mechanisms and significance for research”(brainsci-2730828)
This opinion article advocates the adaptation of the concept of brain reserve for stroke. The proxies, neural mechanisms and significance for research were summarized and proposed. Overall, this topic is interesting and the concept of brain reserve might hold great theoretical and practical implications. However, one concern appeared after reading the whole manuscript.
“To the best of our knowledge, there are no studies assessing the effect of the medial temporal lobe volume or its atrophy on stroke outcome.”this statement seems not so true, please see the systematic review below.
Wang, F., Hua, S., Zhang, Y., Yu, H., Zhang, Z., Zhu, J., ... & Jiang, Z. (2021). Association between small vessel disease markers, medial temporal lobe atrophy and cognitive impairment after stroke: a systematic review and meta-analysis. Journal of Stroke and Cerebrovascular Diseases, 30(1), 105460.
Author Response
“Adaptation of the concept of brain reserve for stroke: proxies, neural mechanisms and significance for research”(brainsci-2730828)
This opinion article advocates the adaptation of the concept of brain reserve for stroke. The proxies, neural mechanisms and significance for research were summarized and proposed. Overall, this topic is interesting and the concept of brain reserve might hold great theoretical and practical implications. However, one concern appeared after reading the whole manuscript.
We appreciate the Reviewer’s assessment of our manuscript and the suggested improvements, which we implemented now.
“To the best of our knowledge, there are no studies assessing the effect of the medial temporal lobe volume or its atrophy on stroke outcome.”this statement seems not so true, please see the systematic review below.
Wang, F., Hua, S., Zhang, Y., Yu, H., Zhang, Z., Zhu, J., ... & Jiang, Z. (2021). Association between small vessel disease markers, medial temporal lobe atrophy and cognitive impairment after stroke: a systematic review and meta-analysis. Journal of Stroke and Cerebrovascular Diseases, 30(1), 105460.
Many thanks for the reference, which we included in the manuscript now. The referred study however reported the association between medial temporal lobe volume and post-stroke cognitive outcome. There are no studies investigating the association between hippocampus volume and non-cognitive stroke outcome. To avoid misinterpretation, we revised the corresponding sentence: P. 5 “To the best of our knowledge, there are no studies assessing the effect of the medial temporal lobe volume or its atrophy on non-cognitive stroke outcome. The negative impact of hippocampus atrophy has been shown in regard to post-stroke cognition 1–4.”

Reviewer 3 Report
Comments and Suggestions for Authors
The manuscript provides insights about challenges associated with predicting stroke outcomes, emphasizing the impact of individual differences. The authors advocated for the application of the “brain reserve” (BR) concept for strokes, although this idea is initially developed for neurodegeneration. Numerous parameters that could be easily monitored on MRI testing, could serve as valuable predictors for stroke outcomes. The authors also suggest that beyond contributing to the understanding of inter-individual variability in stroke outcomes, the BR concept holds promise for personalized treatments and proactive measures in preventing strokes, neurodegeneration, and promoting overall health during the aging process. The minor concern is about the Figure 2 description/legend and incorporation of the description of BR abbreviation in to the legend.
Author Response
The manuscript provides insights about challenges associated with predicting stroke outcomes, emphasizing the impact of individual differences. The authors advocated for the application of the “brain reserve” (BR) concept for strokes, although this idea is initially developed for neurodegeneration. Numerous parameters that could be easily monitored on MRI testing, could serve as valuable predictors for stroke outcomes. The authors also suggest that beyond contributing to the understanding of inter-individual variability in stroke outcomes, the BR concept holds promise for personalized treatments and proactive measures in preventing strokes, neurodegeneration, and promoting overall health during the aging process. The minor concern is about the Figure 2 description/legend and incorporation of the description of BR abbreviation in to the legend.
Many thanks for the appreciation of our work. We revised the legend of Figure 2 now.

Round 2
Reviewer 1 Report
Comments and Suggestions for Authors
Authors did a great job revising the manuscript. Thank you!